# Novel Thin Film Nanocomposite Forward Osmosis Membranes Prepared by Organic Phase Controlled Interfacial Polymerization with Functional Multi-Walled Carbon Nanotubes

**DOI:** 10.3390/membranes11070476

**Published:** 2021-06-28

**Authors:** Xu Zhang, Jiuhan Zheng, Lusheng Xu, Ming Yin, Guoliang Zhang, Wenqian Zhao, Zeyu Zhang, Chong Shen, Qin Meng

**Affiliations:** 1State Key Lab of Green Chemical Synthesis Technology, Center for Membrane and Water Science &Technology, Institute of Oceanic and Environmental Chemical Engineering, Zhejiang University of Technology, Hangzhou 310014, China; xz669592@sina.com (X.Z.); zhengjiuhan@163.com (J.Z.); xulusen@zjut.edu.cn (L.X.); my1757573846@yeah.net (M.Y.); zwq4627@yeah.net (W.Z.); zyzhang11@yeah.net (Z.Z.); 2College of Chemical and Biological Engineering, State Key Laboratory of Chemical Engineering, Zhejiang University, Hangzhou 310027, China; rainbows@zju.edu.cn

**Keywords:** multi-walled carbon nanotubes (MWCNTs), 3-aminopropyltriethoxysilane (APTES), thin film nanocomposite (TFN), interfacial polymerization, forward osmosis (FO)

## Abstract

Novel high-quality thin film nanocomposite (TFN) membranes for enhanced forward osmosis (FO) were first synthesized through organic phase controlled interfacial polymerization by utilizing functional multi-walled carbon nanotubes (MWCNTs). As 3-aminopropyltriethoxysilane (APTES) grafted MWCNTs via an amidation reaction significantly promoted the dispersion in organic solution, MWCNTs-APTES with better compatibility effectively restricted the penetration of trimesoyl chloride (TMC), thus adjusting the morphology and characters of TFN membranes. Various techniques such as Fourier transform infrared spectra (FTIR), transmission electron microscopy (TEM), X-ray photoelectron spectroscopy (XPS), scanning electron microscope (SEM), sessile droplet analysis and FO experiments and reverse osmosis (RO) operation were taken to characterize and evaluate the performance of nanocomposites and membranes. The prepared TFN FO membranes exhibited good hydrophilicity and separation efficiency, in which water flux was about twice those of thin film composite (TFC) membranes without MWCNTs-APTES in both AL-DS and AL-FS modes. Compared with the original TFC membrane, the membrane structural parameter of the novel TFN FO membrane sharply was cut down to 60.7%. Based on the large number of low mass-transfer resistance channels provided by functional nanocomposites, the progresses may provide a facile approach to fabricate novel TFN FO membranes with advanced selectivity and permeability.

## 1. Introduction

With continuous development of the world’s population and serious environmental pollution, water resources are far from being adequate to meet the needs of human beings for drinking and freshwater shortages have become a serious problem in the world [1,2]. In order to mitigate the water crisis, scientists have focused on using membrane separation technology to desalination recently [3,4,5,6,7]. Among various membrane desalination technologies, the osmotic pressure driven forward osmosis (FO) technology has attracted extensive attention [8,9]. Different from reverse osmosis (RO) with external pressure driven, osmotic pressure is the driving force of FO process, in which the pure water of the low osmotic pressure solution is spontaneously diffused to the high osmotic pressure solution. Therefore, FO may break the limitations of RO for its low energy consumption, higher pollution reversibility and higher water reuse, and is expected to become a new type of seawater desalination technology to alleviate water and energy crisis [10,11,12].

Nevertheless, a major challenge that impedes FO technology application is the shortage of FO membranes with high separating property [13]. Among all kinds of FO membranes, thin film composite (TFC) membranes containing polyamide (PA) active layers on the support layer is regarded as the state-of-art FO membrane, due to that TFC membranes still have desirable separation ability over a wide range of operating temperature and pH and can independently optimize the support and selective layers [14,15]. However, the relative low water permeability, material fouling, harsh internal concentration polarization (ICP) and undesirable reverse solute flux of TFC membranes restricted the applications of the TFC FO membrane [16,17]. Therefore, to achieve the optimal separation performance, the chemical and structure properties of TFC FO membrane must be intensively designed and greatly enhanced.

With the rapid development of nanotechnology, incorporating nanomaterials into the FO membrane may be a practical strategy to enhance the separating property of membranes [18,19,20,21]. In this respect, there are usually two ways. One is to increase the support layer capability of the TFC membrane, for example, optimizing the pore structure, boosting hydrophilicity and porosity and thus mitigating the ICP of the FO membrane. Another way is to enhance the PA active layer performance of the FO membrane. The concept of thin film nanocomposite (TFN) membrane formed by incorporating nanofillers into the PA layer was proposed. Up to now, a wide variety of nanomaterials, such as graphene oxide, metal/metal oxide and molecular sieve nanoparticles and so on, have been used as a filler to fabricate TFN membranes [22,23,24,25,26,27]. Among them, multi-walled carbon nanotubes (MWCNTs) were focused on modifying the TFC membrane for its excellent stability, high mechanical property and intrinsically remarkable transport channels. With embedded MWCNTs in the active layer of the TFN membrane, the water flux and antifouling capability can be improved [28]. However, because of their large aspect ratio and high surface energy, the compatibility between MWCNTs and PA active layer was always unsatisfactory, leading to localized defects and the damaged integrity of the PA active layer of FO membrane [29,30,31]. Moreover, the hydrophilicity of MWCNTs was poor, which turned out to be bad for the flux improvement of the FO membrane. Considering that the affinity with the polymeric matrix and the dispersion in the solution of nanofillers can be effectively improved by applying the appropriate modification strategy [28,32], we expect that exploration of modified MWCNTs to prepare the TFN membrane may be an effective way to break through the current bottlenecks.

Generally, since the nanofillers display better dispersibility in aqueous solution, TFN membranes can be prepared by adding nanofillers into the aqueous solution [33,34]. When nanofillers are incorporated to the aqueous solution, most of the nanofillers will be lost from the membrane surface with the removal of excess aqueous solution, greatly reducing the load of nanofillers and weakens its excellent performance [35]. Comparatively, with adding nanofillers into the organic solution, most of the nanofillers will remain because the nanofillers are embedded in the PA layer and can hardly be removed from the organic solution. In this view, since MWCNTs have one-dimensional nanostructures with large open channels of more than 5 nm, incorporation of MWCNTs into the active layer of the TFN membrane may reduce the mass transfer resistance to a great extent. Even if taking account of the characteristics of MWCNTs, the modification of MWCN.

Ts can obviously improve the hydrophilicity and dispersity of the MWCNTs in organic solution, and the new type of material compatibility between the MWCNTs and PA separation layer will be achieved. Although nanofillers such as metal organic frameworks (MOFs) and microporous carbon have been attempted to prepare the TFN FO membrane by incorporating them in the organic phase of interfacial polymerization [26,36] and the as-synthesized TFN membrane showed good water permeability and, so far, there is no report on fabricating the TFN FO membrane by adding MWCNTs, especially modified MWCNTs, as efficient nanofillers in the organic phase during the interfacial polymerization process for enhanced forward osmosis.

Based on the above considerations, in this study, we propose a facile and low-cost method of the organic phase controlled interfacial polymerization to synthesize the novel TFN membrane for enhanced forward osmosis by applying 3-aminopropyltriethoxysilane (APTES) grafted MWCNTs (Figure 1). Our strategy has the following advantages. First, grafting APTES (MWCNTs-APTES) to the MWCNTs promoted their dispersion in organic solution. Second, MWCNTs-APTES with better compatibility towards TMC sharply restricted the penetration of TMC, making the favorable morphology of the TFN FO membrane. Third, the MWCNTs-APTES contained hydrophilic amide groups, a carboxyl group and ethoxy silane, which can hydrolyze into the hydrophilic hydroxyl group and thus increased the hydrophilicity of the resulted TFN FO membranes. As far as we know, it is the first time exploring APTES modified MWCNTs to fabricate a high-quality TFN FO membrane. The resulted TFN membranes made by this strategy displayed considerable FO desalination behaviors due to the hydrophilic groups and intrinsic mass transfer channels of MWCNTs-APTES.

## 2. Materials and Methods

### 2.1. Materials

Multi-walled carbon nanotubes (MWCNTs) were supplied by Nanotech Port Co., Ltd. (Nanjing, China), with an average length of greater than 5 µm and the average diameter of 10–20 nm. The polysulfone (PSF) particle (molecular weight of 65,000 Da) was purchased from Dalian Polysulfone Plastic Co., Ltd. (Dalian, China). Concentrated sulfuric acid, nitric acid and tetrahydrofuran were purchased from Lingfeng chemical reagent co., Ltd. (Shanghai, China). N,N-dimethyl formamide (DMF), HNO_3_, H_2_SO_4_, methanol, ethanol, thionyl chloride (SOCl_2_) and n-hexane were supplied by Sinopharm Chemical Reagent Co., Ltd. (Hangzhou, China). and were not purified before being used. Trimesoyl chloride (TMC), m-phenyldiamine (MPD) and 3-aminopropyltriethoxysilane (APTES) were supplied by Aladdin Chemical Co., Ltd. (Shanghai, China). The deionized (DI) water was made through the RO-EDI high purification system.

### 2.2. Preparation of MWCNTs-APTES

The synthesis mechanism of MWCNTs-APTES was shown in Figure 1. The carboxyl-modified MWCNTs were first prepared similar to that reported elsewhere [37]. Specifically, the pristine MWCNTs were added into the mixed solution of concentrated HNO_3_ and H_2_SO_4_ (*v*/*v*: 1:3) and was followed by refluxing at 80 °C. Subsequently, the resulted samples were washed with DI water until the water pH was about 7 and then was dried. The as-synthesized MWCNTs with oxidized carboxylic groups on the outer walls were referred to as MWCNTs-COOH. Then, MWCNTs-COOH were mixed with SOCl_2_ and DMF and a refluxed reaction at 70 °C occurred, the products were washed with tetrahydrofuran and dried (referred as MWCNTs-COCl). MWCNTs-COCl was added into APTES and a refluxed reaction at 80 °C occurred. The final products MWCNTs-APTES was washed with DMF and methanol washing and dried.

### 2.3. Preparation of the PSF Support Layer

A total of 12 wt % of PSF particles were soluted in DMF at room temperature. Subsequently, the resulting casting solution was treated with ultrasonic vibrations for 30 min to remove air bubbles. The resultant solution was cast on the glass substrate and then the coated glass was immediately immersed in the DI water coagulation bath. The obtained PSF substrate was kept in DI water for at least one day before being used.

### 2.4. Fabrication of TFC and TFN FO Membranes

The TFC and TFN FO membranes were produced by interfacial polymerization on PSF substrates. The upper surface of the membrane was contacted with the organic/aqueous phase solution. In detail, the PSF substrate membrane was first exposed to the MPD (1% *w*/*v*) solution. Accordingly, the MPD-soaked PSF substrate was immersed in the n-hexane solution containing TMC (0.05% *w*/*v*) and MWCNTs-APTES (0.00%, 0.05%, 0.10%, 0.20% and 0.40% *w*/*v*). The as-synthesized membrane was placed at 60 °C. The prepared FO membranes were completely rinsed and stored in DI water before being used. These FO membranes were noted as TFC and TFN-x, where ‘x’ is the loading content of MWCNTs-APTES.

### 2.5. Characterizations

Fourier transform infrared spectra (FTIR, Nicolet 6700, Thermo Fisher Scientific, Waltham, MA, USA) and X-ray photoelectron spectroscopy (XPS, Thermo Scientific Inc., London, UK) was used to test the chemical structure of the samples. The weight loss of MWCNT-based materials was determined using thermogravimetric analysis (TGA, Pyris1TGA, PerkinElmer, Waltham, MA, USA). Scanning electron microscope (SEM, SU-70, Hitachi, Tokyo, Japan) transmission electron microscopy (TEM, Tecnai G2 F30 S-Twin, Amsterdam, The Netherlands) were used to examine the sample morphology. The water contact angles (WCAs) of membranes were gauged by the contact angle meter (OCA20). In order to reduce the error of measurement, the test was at a different location test for at least five times.

### 2.6. Determination of Membranes Intrinsic Separation Performance

The pure water permeability (A) and salt rejection (R) of the as-synthesized membranes were assessed at 6 bar by a cross-flow RO setup (Appendix A). The effective membrane area was 7.065 cm^2^. A 20 mM NaCl solution was employed as the feed solution. A, R and the salt permeability coefficient (B) of the TFC and TFN membranes were obtained by applying Equations (1)–(3).
(1)$$A=\frac{J}{\Delta P}\;$$
(2)$$R(\%)=(1-\frac{C_{p}}{C_{f}})\times 100\;$$
(3)$$\frac{1-R}{R}=\frac{B}{A\left(\Delta P-∆\pi \right)}\;$$
where J represents the water flux. C_f_ and C_p_ are the feed concentration and permeate concentration, respectively, while ΔP and Δπ represent the actuating operation pressure and osmotic pressure across the FO membrane respectively.

### 2.7. Estimation of TFC and TFN Membranes’ FO Performance

FO performances of the resulted membranes were valued by homemade cross-flow FO equipment (practical membrane area of 3.00 cm^2^). The cross-flow rate was 5.0 m min^−1^. DI water and 1 M NaCl solution were the feed and draw solution, respectively. The membrane performance was assessed under both AL-FS (feed solution on the PA selective layer side) and AL-DS (draw solution on the PA selective layer side) modes.

The FO water flux (J_V_) was determined according to Equation (4).
(4)$$J_{V}=\frac{\Delta m_{\mathrm{feed}}}{\rho_{\mathrm{feed}}\times A_{m}\times \Delta_{t}}\;$$
where Δm_feed_, ρ_feed_, A_m_ and Δ_t_ are the mass change of the feed solution, the density of water, the effective membrane area and the operation time, respectively. There solute flux J_S_ was obtained according to Equation (5).
(5)$$J_{S}=\frac{\Delta \left(C_{t}{\times V}_{t}\right)}{A_{m}\times \Delta t}$$
where V_t_ and C_t_ stand for the volume and solute concentration of the feed solution, respectively.

According to the traditional ICP model [38], FO water flux of the membrane under AL-FS and AL-DS modes are computed from Equations (6) and (7) respectively, consequently, the membrane structural parameter (S) can be procured.

AL-FS mode:(6)$$J_{V}=\frac{D}{S}\left[\ln\frac{{A\pi }_{\mathrm{draw}}+B}{{A\pi }_{\mathrm{feed}}+J_{V}+B}\right]$$

AL-DS mode:(7)$$J_{V}=\frac{D}{S}\left[\ln\frac{{A\pi }_{\mathrm{draw}}-J_{V}+B}{{A\pi }_{\mathrm{feed}}+B}\right]$$
where D represents the solute diffusion coefficient, π_feed_ and π_draw_ are the feed and draw solution osmotic pressures, respectively.

## 3. Results and Discussion

### 3.1. Characterization of MWCNTs-APTES

FTIR spectroscopy was used to analyze the chemical structure of the MWCNTs-based materials. Figure 2 showed the FTIR result of original MWCNTs, MWCNTs-COOH, MWCNTs-COCl and MWCNTs-APTES. For the untreated MWCNTs, the peak appearing at 2800–2980 cm^−1^ accorded with the C-H stretching vibration in methylene, the band range of 1400–1600 cm^−1^ was the feature absorption peak of aromatic ring, the range of 700–750 cm^−1^ and 850–900 cm^−1^ corresponded to the out of plane bending vibration of the aromatic ring. The new characteristic peak of MWCNTs-COOH appeared at 1700–1750 cm^−1^, which was caused by the C=O stretching vibration, proving that mixed acid successfully oxidized MWCNTs to bring the carboxyl group on the surface [39]. After acyl chlorination, a new characteristic peak of MWCNTs-COCl appeared at 1785–1815 cm^−1^, which was caused by the acyl chloride group stretching vibration. After further grafting APTES, a broad peak appeared between 3400 and 3500 cm^−1^ originated from the stretching vibration of the N-H bond. In addition, the characteristic peaks of 1370–1400 cm^−1^ and 1050–1100 cm^−1^ corresponded to stretching and bending vibrations of C-N and Si-O-C respectively, and the characteristic peaks of acyl chloride groups disappeared, indicating that APTES were favorably grafted onto the surface of MWCNTS by a covalent bond [40]. TEM was then applied to observe the MWCNTs-APTES (Figure 3). The TEM images suggest that the overall structure of MWCNTs was not damaged after grafting APTES and the outer surface of MWCNTS-APTES had more amorphous structure, proving that the APTES was successfully grafted onto the out surface of MWCNTs. The TGA test of pure MWCNTs, MWCNTs-COOH, MWCNTs-COCl and CNTs-APTES showed that the mass of MWCNTs had almost no change when the temperature was below 500 °C (Appendix A). However, compared with the original MWCNTs, the mass loss of MWCNTs-COOH and MWCNTs-APTES was obvious, which was due to the carboxyl and organosilane functional groups respectively, further demonstrating that the MWCNTs had been modified successfully.

To further analyze the functional groups of the MWCNTs-based materials, XPS analysis was conducted. As shown in Figure 4, it can be found that the MWCNTs-COOH, MWCNTs-COCl and MWCNTs-APTES primarily comprised carbon and oxygen elements. After acyl chlorination, a new peak appeared at 201.4 eV, which was the characteristic peak of Cl 2p^3^. After treating MWCNTs-COCl with APTES, the characteristic peaks corresponding to Si 2p^3^ and N 1s appeared at bond energies of 104 eV and 398.4 eV [40], respectively, indicating that APTES had been successfully grafted onto the MWCNTs-COCl, which was consistent with the results of the FTIR spectra analysis.

C 1s and N 1s spectrum were analyzed to further investigate the reaction principle of the MWCNTs modification approach (Figure 5). The C1s spectrum of MWCNTs-COOH (Figure 5a) was resolved into C-C=C (284.6 eV), C-OH (286.2 eV) and O-C=O (289.4 eV), indicating that the surface of carbon nanotubes after oxidation contained hydroxyl and carboxyl functional groups. After the acyl chlorination reaction (Figure 5b), a characteristic peak belonging to C-Cl (286.4 eV) was detected, the peak of the -O-C=O bond disappeared and was transformed into the peak of C=O, confirming the transformation of carboxyl groups on MWCNTs-COOH to acyl chloride. From the XPS C 1s nuclear spectrum of MWCNTs-APTES (Figure 5c), a peak of the C-N bond appeared at 286.2 eV, and the C=O still existed, while the C-Cl disappeared. This indicates that the acyl chloride group (-COCl) on MWCNTs-COCl reacted with the amino (-NH_2_) of APTES, which are grafted to MWCNTs by covalent bonds. The N 1s XPS result of MWCNTs-APTES also confirmed the above conclusion. For MWCNTs-APTES (Figure 5d), three peaks corresponding to C-N (398.2 eV), O=C-N (299.8 eV) and C-N^+^H (401.4 eV) could be observed, therefore, it can be concluded that APTES was grafted onto MWCNTs-COCl through a covalent bond.

The dispersity of MWCNTs and MWCNTs-APTES in the n-hexane solution was tested (Appendix A). After 10 min of the ultrasonic dispersion treatment, it was found that both MWCNTs and MWCNTs-APTES were uniformly dispersed in the n-hexane solution. However, after the two dispersions were stand for only 30 min, it was found that MWCNTs had precipitated and the supernatant was almost completely transparent, indicating that MWCNTs had almost completely precipitated. For MWCNTs-APTES, almost no particle precipitation was found. The above results showed that the dispersity of MWCNTs-APTES in n-hexane was obviously better than that of unmodified MWCNTs, indicating that the APTES chain segment promoted the dispersion of MWCNTs in the n-hexane solution.

### 3.2. Characterization of TFC and TFN Membranes

The membrane morphologies were displayed by using SEM. From Figure 6a,b, the PSF membrane had large pore structure and spongy pore structure, and the large opening structure was found at the bottom, which helped to reduce the ICP in the FO process. In detail, the upper surface of the PSF membrane showed pores with a diameter of 50–100 nm and thus the water flux of the fabricated PSF membrane was relatively large (322.7 L m^−2^ h^−1^ bar^−1^). The resulted PSF membrane was applied as a support layer to synthesize TFC and TFN membranes via the IP process. As is shown in Figure 6, it is visible that the TFC membrane displayed typical “ridge and valley” morphology owing to the IP reaction between TMC and MPD. However, after incorporating MWCNTs-APTES, the PA dense layer morphologies of the resulted TFN membranes were remarkably different from that of TFC membranes and turned to the “leaf-like” structure. The result may be attributed to the MWCNTs-APTES that participated in the IP reaction. On the other hand, the presence of MWCNTs-APTES hindered the diffusion of MPD and slowed down the PA formation, resulting in a loose surface.

The surface hydrophilicity of the PSF membrane, TFC membrane and TFN membrane was characterized by estimating the WCA of the membrane (Figure 7). For the PSF substrate, the WCA value was measured as 83.7°. After preparing the PA layer on the PSF support, the WCA value of the TFC membrane increased to 95.9°, which was due to the network structure containing a large number of benzene rings through the cross-linking of TMC and MPD, making the surface of the TFC membrane more hydrophobic. With introducing MWCNTs-APTES into the PA active layer, the WCA of TFN membranes were initially decreased. When the loading amount of MWCNTs-APTES increased to 0.2% (*w*/*v*), the WCA of the TFN membrane was the lowest and reduced to 33.1°. It can be attributed to that of the amide group and the remaining carboxyl group of the MWCNTs-APTES were hydrophilic and the ethoxy silane on the MWCNTs-APTES was hydrolyzed into hydrophilic hydroxyl, which made the TFN membranes have better hydrophilicity. Meanwhile, when the loading amount of MWCNTs-APTES amplified to 0.4% (*w*/*v*), the WCA of the TFN membrane slightly increased. The reason may be that in the synthetic process of the PA layer, too many MWCNTs-APTES tended to agglomerate and heaped up on the TFN membrane surface, giving rise to the increased roughness of TFN membranes and reducing the hydrophilicity of TFN membranes.

### 3.3. Intrinsic Separation Properties of TFC and TFN Membranes

The interior separation capabilities of synthesized TFC and TFN membranes were measured by homemade cross-flow filtration equipment and the results were displayed in Table 1. Generally, a good FO membrane should possess high A and small B. The A value of the TFC membrane was relatively low (1.56 L m^−2^ h^−1^ bar^−1^). After incorporating MWCNTs-APTES into the PA layer, the A value of the prepared TFN membrane increased to 4.1 L m^−2^ h^−1^ bar^−1^, which was about 2.6 times larger than that of the TFC membrane. Even though the loading amount of MWCNTs-APTES was as low as 0.05% (*w*/*v*), the A value of TFN membrane reached 3.11 L m^−2^ h^−1^ bar^−1^, which was almost 2.0 times higher than that of the TFC membrane. The increase of the A value of TFN membranes can be ascribed to the adding of MWCNTs-APTES. The diameter of MWCNTs-APTES is about 10–20 nm, providing unobstructed channels of water molecules and increasing the TFN membranes’ pure water flux. In addition, the Si-OH from the hydrolysis of ethoxy-silane groups of MWCNTs-APTES led to improve the hydrophilicity of the TFN membrane. Similarly, the water flux decline of the TFN-0.4 membrane may ascribe to the poorer hydrophilicity of this membrane. The poorer hydrophilicity of the TFN membrane restricted H_2_O molecules into the polyamide matrix and was bad for H_2_O molecule transport through the membrane.

Besides the water penetrability, the rejection of NaCl by TFC and TFN membranes were also shown in Table 1. Compared with the TFC membrane (NaCl rejection was 96.1%), the NaCl rejection of the TFN membrane firstly increased with incorporating MWCNTs-APTES and reached 97.4%. Correspondingly, the B/A value represented the separation performance of the TFN membrane and was smaller than that of the TFC membrane, exhibiting that the separation efficiency of the resulted TFN membrane was better than that of the TFC membrane. However, when the loading amount of MWCNTs-APTES was further increased, the separation ability of the TFN membrane was the worst in NaCl rejection on account of the agglomeration of MWCNTs-APTES and the formed nonselective defects.

### 3.4. FO Performance

FO separation performance of the fabricated TFC and TFN FO membranes were measured under both AL-DS and AL-FS modes, respectively. From Figure 8a,b, with increasing the loading amount of MWCNTs-APTES, the water flux of the resulted FO membrane presented a similar trend with the pure water flux (Table 1). Moreover, compared with the TFC membrane, it is obvious that TFN membranes modified with MWCNTs-APTES had higher water flux under the two operation modes, because of the loose surface structure and abundant oxygen-containing functional groups of the TFN membranes. For the TFC membrane, J_V_ was 12.6 in the AL-DS mode and 8.2 L m^−2^ h^−1^ in the AL-FS mode. When the loading amount of MWCNTs-APTES increased to 0.2% (*w*/*v*), under AL-DS and AL-FS modes, J_V_ of the TFN membrane were 22.6 and 19.6 L m^−2^ h^−1^, about 80% and 140% larger than that of the original TFC membrane, respectively. The improvement of J_V_ can be on account of the lower permeation resistance with introducing MWCNTs-APTES into the PA active layer. It is worth taking into account that the J_V_ of TFN membranes had no significant changes with further increases in the nanofiller amount. For J_S_, when the loading amount of MWCNTs-APTES was lower than 0.1% (*w*/*v*), J_S_ increased slightly. However, a further increase of the MWCNTs-APTES loading amount resulted in a sharp increase in J_S_, which was due to the MWCNTs-APTES agglomerates in TFN membranes.

As noted, the concentrative ICP impact can be negligible in the AL-DS mode. Nevertheless, the dilutive ICP impact is more serious in the AL-FS mode, because the osmotic pressure decreases as the operation is prolonging and thus decreases the water flux to a great extent [41,42]. This is why the water flux under the AL-FS mode is always less than that of the AL-DS mode. As shown in Figure 8, for all membranes, J_V_ in the AL-DS direction was greater than that in the AL-FS direction, which can be ascribed to the better ICP in this direction. In addition, in the AL-FS mode, the water flux increased more remarkably. Apart from the effect of MWCNTs-APTES on TFN membrane morphology, MWCNTs-APTES contained the hydrophilic group and intrinsic mass transfer channels greatly promoted the infiltration of H_2_O molecules into the TFN membrane, which was conducive to the rapid passage of H_2_O molecules through the TFN membrane and the alleviation of the ICP phenomenon for the TFN membrane. S is a vital index to estimate the ICP impact of the FO membrane and was calculated from the FO and RO results. In general, the smaller the S value is, the lower the ICP impact is, with increasing FO flux. Compared to the TFC membrane, the S value of the TFN membrane with 0.4% (*w*/*v*) MWCNTs-APTES loading dropped down to 610 μm, sharply cutting down to 60.7%. The results indicated that the ICP effect of the fabricated TFN membranes were mitigated after incorporating modified MWCNTs to some extent. In addition, J_S_/J_V_ was evaluated to assess the selectivity of the FO membrane. From Figure 8c,d, with the increase of the MWCNTs-APTES loading amount, the J_S_/J_V_ value of the fabricated TFN membrane firstly decreased and subsequently increased. When the loading of MWCNTs-APTES was 0.1% (*w*/*v*), the lowest J_S_/J_V_ was achieved, which was reduced to 0.091 g L^−1^ (AL-FS mode), demonstrating that the selectivity of the TFN membrane was greatly improved by incorporating MWCNTs-APTES. Compared with other CNT-based FO membranes (Appendix A), the J_S_/J_V_ of the resulted TFN membrane was also very low, indicating that incorporating MWCNTs-APTES into the TMC solution during interfacial polymerization could effectively enhance the FO performance of the TFN membrane.

## 4. Conclusions

In summary, novel TFN FO membranes with mitigating internal concentration polarization were successfully synthesized by utilizing APTES modified MWCNTs via the organic phase controlled interfacial polymerization strategy. Through the amidation chemical reaction, APTES was first successfully grafted onto MWCNTs, and the resulted MWCNTs-APTES showed better dispersity in the n-hexane solution compared with the original MWCNTs. After introducing MWCNTs-APTES into the PA layer, the pure water flux of the TFN membrane was improved to a great extent. When the addition amount of MWCNTs-APTES was 0.1% (*w*/*v*), the permeation flux of the TFN membrane increased to 140% and reached 22.6 L m^−2^ h^−1^ (AL-DS mode). Moreover, the J_S_/J_V_ value of the TFN membrane reduced to 0.091 g L^−1^ (AL-FS mode). Compared with the original TFC membrane, the membrane structural parameter of the novel TFN FO membrane sharply cut down to 60.7%. The progresses in this research provide a new way for the production of high-quality thin film forward osmosis membranes by applying incorporated functional nanocomposites, which displays great potential in a widely industrial application.

## Figures and Tables

**Figure 1 membranes-11-00476-f001:**
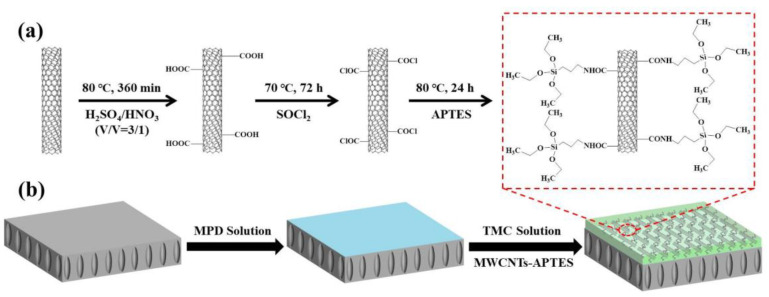
Schematic illustration of MWCNTs grafted with APTES (**a**) and TFN membrane synthesis (**b**).

**Figure 2 membranes-11-00476-f002:**
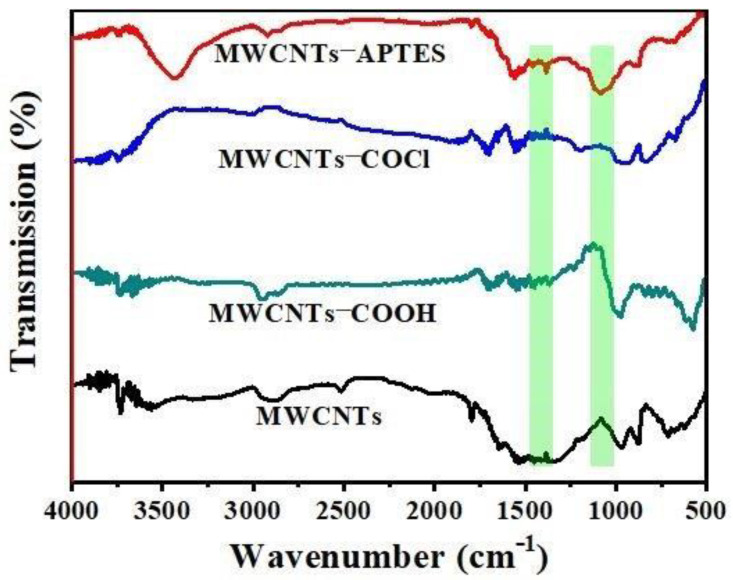
FTIR spectra of MWCNTs, MWCNTs-COOH, MWCNTs-COCl and MWCNTs-APTES.

**Figure 3 membranes-11-00476-f003:**
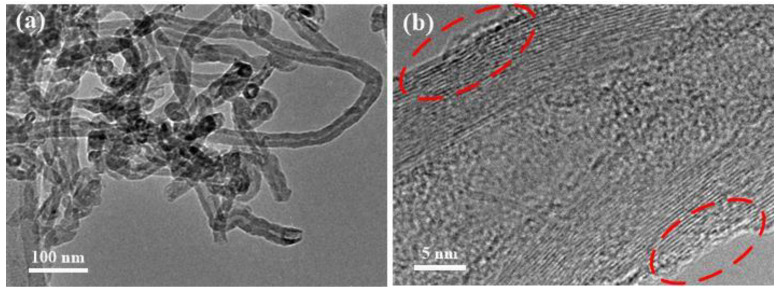
TEM image of MWCNTs-APTES (**a**) low-magnification microscope and (**b**) high-magnification microscope.

**Figure 4 membranes-11-00476-f004:**
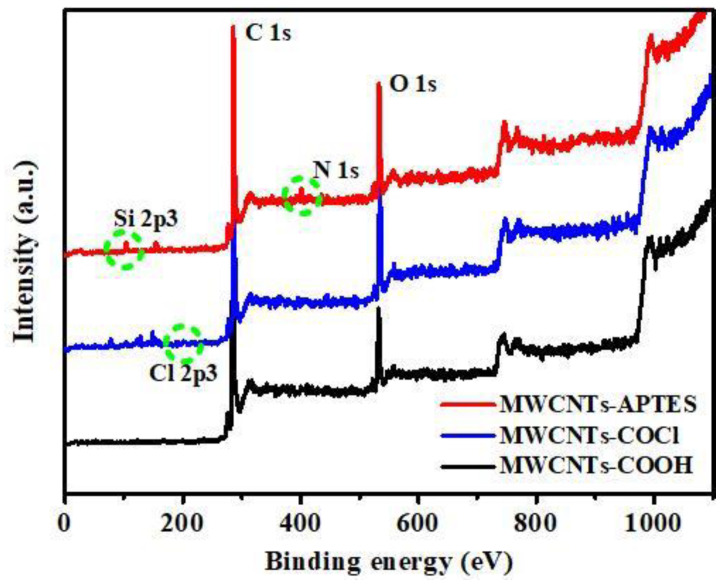
Wide-scan survey XPS spectra of MWCNTs, MWCNTs-COOH, MWCNTs-COCl and MWCNTs-APTES.

**Figure 5 membranes-11-00476-f005:**
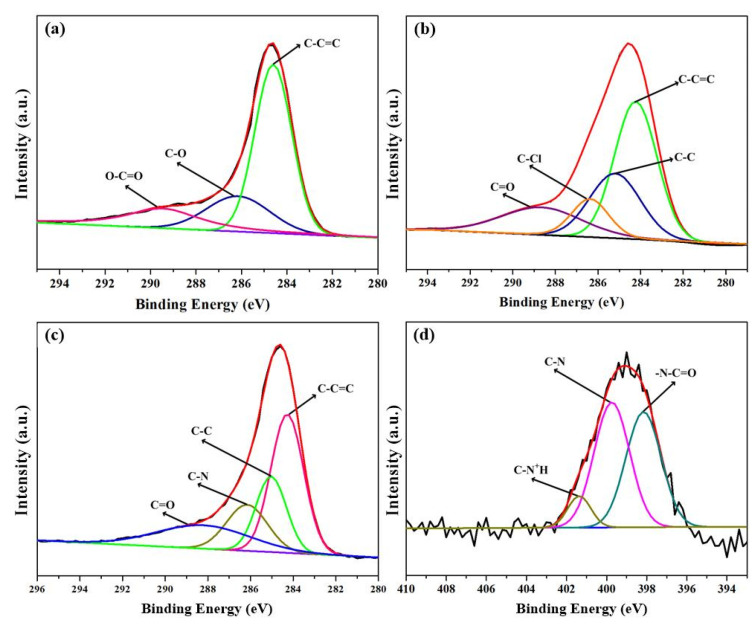
The C 1s spectrum of MWCNTs-COOH (**a**), MWCNTs-COCl (**b**) and MWCNTs-APTES (**c**) and N 1s spectrum of MWCNTs-APTES (**d**).

**Figure 6 membranes-11-00476-f006:**
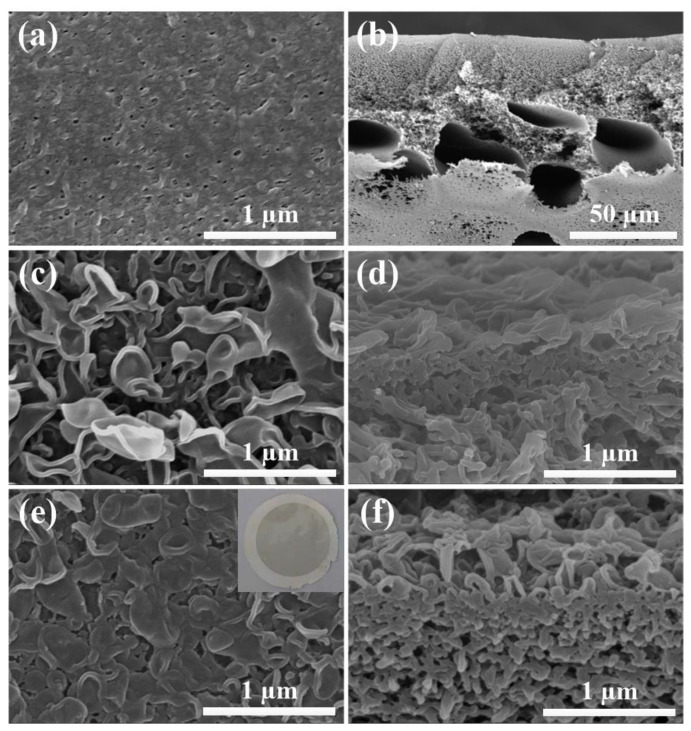
SEM images of surfaces and the cross-section for the PSF membrane (**a**,**b**), TFC membrane (**c**,**d**) and TFN-0.1 membrane (**e**,**f**).

**Figure 7 membranes-11-00476-f007:**
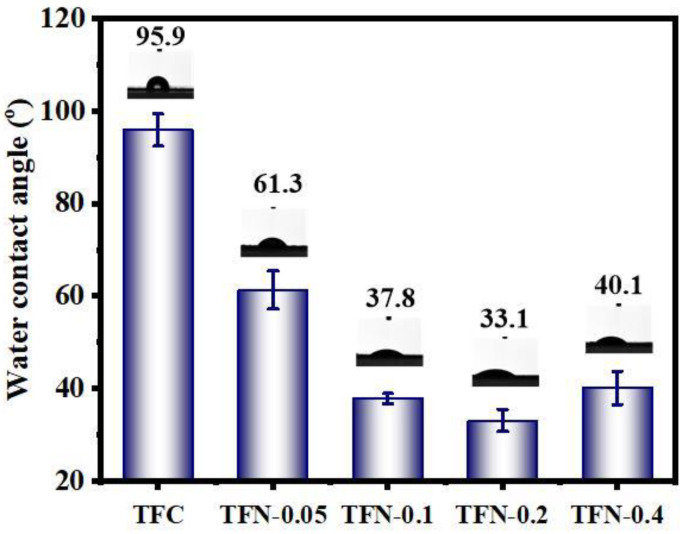
Water contact angle of TFC and TFN membranes.

**Figure 8 membranes-11-00476-f008:**
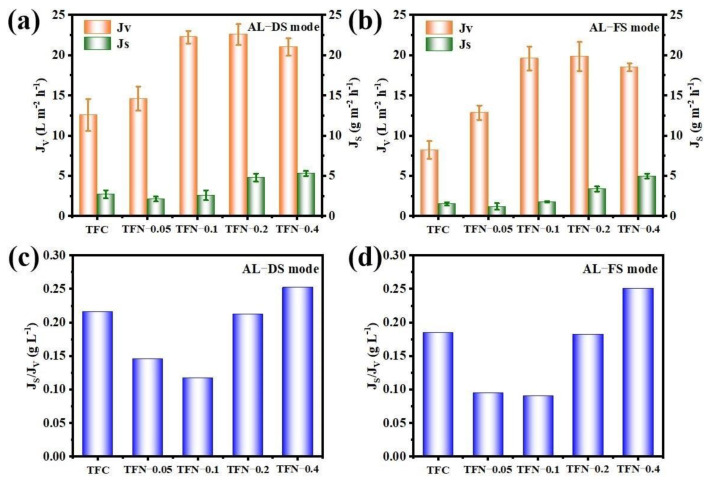
J_V_, J_S_ and J_S_/J_V_ of TFC and TFN membranes in FO applications under both AL-DS (**a**,**c**) and AL-FS (**b**,**d**) modes.

**Table 1 membranes-11-00476-t001:** Salt rejections (R), intrinsic properties and membrane structural parameters (S) of the TFC and TFN membranes.

Membrane	Rejection of NaCl (%)	A (L m^−2^ h^−1^ bar^−1^)	B (L m^−2^ h^−1^)	B/A (Bar)	S (µm)
TFC	96.1 ± 0.76	1.56 ± 0.08	0.32 ± 0.02	0.205 ± 0.002	1555 ± 93
TFN-0.05	97.3 ± 0.65	3.11 ± 0.03	0.40 ± 0.01	0.129 ± 0.001	1114 ± 64
TFN-0.1	97.4 ± 1.16	3.59 ± 0.12	0.48 ± 0.02	0.134 ± 0.007	646 ± 18
TFN-0.2	95.7 ± 0.84	4.10 ± 0.09	0.92 ± 0.07	0.224 ± 0.013	737 ± 47
TFN-0.4	93.6 ± 0.53	3.40 ± 0.16	1.16 ± 0.05	0.341 ± 0.012	610 ± 32

## Data Availability

Data is contained within the article or Appendix A.

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
