# Peer review of "Novel Thin Film Nanocomposite Forward Osmosis Membranes Prepared by Organic Phase Controlled Interfacial Polymerization with Functional Multi-Walled Carbon Nanotubes"

_membranes, 2021, doi:10.3390/membranes11070476_

Round 1

Reviewer 1 Report

see attached file

Author Response

Reviewers’ comments:

Referee 1:

The authors report on the functionalization of multiwall carbon nanotubes (MWCNT) and their integration via the organic phase during interfacial polymerization into the barrier layer of thin-film composite polyamide membranes. The evaluation under RO and FO conditions indicated that this approach can lead to improved separation performance. The conclusions are based on multi-method characterizations according to the state-of-the-art. Overall, the study is well performed and the paper is of good formal quality.

I have only a few comments, most just on small formal aspects (listed in the order of appearance):

  1. All abbreviations in Abstract must be explained; if the term is used only once in Abstract, no abbreviation is necessary there.

Answer: Thank you for your suggestion. All abbreviations in Abstract have been explained.

  1. 60: “ICP” must be explained

Answer: In fact, we have explained “ICP” at line 52.

  1. 105: “silicon ethoxy” should be change to correct chemical nomenclature “ethoxy silane”

Answer: Thank you for your suggestions. We have corrected the nomenclature in the revised manuscripts.

  1. 132: “SOCl2” … “2” must be subscript

Answer: Thank you for your suggestions. We have subscripted it in the revised manuscripts.

  1. 173: giving cross-flow rare in “mL min-1” is meaning-less; the linear velocity (in “m min-1”) is relevant to judge about efficiency; providing the Reynolds number would be even better.

Answer: Thank you for your suggestions. The linear cross-flow rate was 5.0 m min-1.

  1. 196: Do not use “FTIR” alone; here it should read “FTIR spectroscopy”

Answer: Thank you for your suggestions. We have corrected it in the revised manuscripts.

  1. 219: “resoundingly” seems to be the wrong word; I do not know what they wish to express here.

Answer: Thank you for your suggestions. We have corrected it as “successfully” in the revised manuscripts.

  1. 229: change “XPS testing” to “XPS analysis”

Answer: Thank you for your suggestions. We have revised it in the revised manuscripts.

  1. 238: change to “C1s spectrum”

Answer: Thank you for your suggestions. We have revised it in the revised manuscripts.

  1. 268: “membrane showed a nanometer hole with a pore diameter of 50-100 nm” should be changed to “membrane showed pores with a diameter of 50-100 nm”

Answer: Thank you for your suggestions. We have revised it in the revised manuscripts.

  1. 290: “silicon ethoxy” should be change to correct chemical nomenclature “ethoxy silane”

Answer: Thank you for your suggestions. We have revised it in the revised manuscripts.

  1. 306: it should read “2.6 times larger”

Answer: Thank you for your suggestions. We have revised it in the revised manuscripts.

  1. 308: it should read “2 times higher”

Answer: Thank you for your suggestions. We have revised it in the revised manuscripts.

  1. 310-11: MWCNT “providing unobstructed channels of water molecules and increasing the TFN’ membranes’ pure water flux” … This is a sheer speculation. And it also does not make sense, because when water would flow through the about 5 nm wide channels of the MWCNT, salt would also pass through the membrane. However, later the authors report that salt rejection can even increase with addition of MWCNT, for membranes that show higher water permeance.

Answer: Thank you for your comments. Generally, for TFN membrane, the salt rejection was depended on the polyamide active layer to a great extent. In this work, the presence of MWCNTs-APTES can affect the diffusion of MPD, resulting in less defects and higher cross-linking degree of the PA layer. Thus, the salt rejection can even increase with addition of MWCNT.

  1. 314: The authors should explain why “poorer hydrophilicity” would cause lower water flux (compared to other membranes). I know that this is often written (without pointing to a reasonable mechanism). However, the water permeance is determined by the barrier layer and now its surface properties.

Answer: Thank you for your suggestions. The poorer hydrophilicity of TFN membrane restricted H2O molecules into the polyamide matrix and was bad for H2O molecules transport through the membrane. In addition, the aggregation of MWCNTs-APTES contributed to a relatively dense polyamide active layer. Thus, the water flux decreased.

  1. 344: Provide also errors for measured data presented in Table 1.

Answer: Thank you for your suggestions. We have added the errors for measured data in Table 1.

Table 1 Salt rejections (R), intrinsic properties and membrane structural parameters (S) of the TFC and TFN membranes.

Membrane

Rejection of NaCl (%)

A

(L m-2 h-1 bar-1)

B

(L m-2 h-1)

B/A

(Bar)

S

(µm)

TFC

96.1±0.76

1.56±0.08

0.32±0.02

0.205±0.002

1555±93

TFN-0.05

97.3±0.65

3.11±0.03

0.40±0.01

0.129±0.001

1114±64

TFN-0.1

97.4±1.16

3.59±0.12

0.48±0.02

0.134±0.007

646±18

TFN-0.2

95.7±0.84

4.10±0.09

0.92±0.07

0.224±0.013

737±47

TFN-0.4

93.6±0.53

3.40±0.16

1.16±0.05

0.341±0.012

610±32

  1. 347: should read “decreases as the operation is prolonging”

Answer: Thank you for your suggestions. We have revised it in the revised manuscripts.

Thank you for all your comments and suggestions!

06/16/2021

Reviewer 2 Report

The present work describes a method (i.e. functionalization with aminopropyl triethoxy silane) to efficiently incorporate multi-walled carbon nanotubes to thin film forward osmosis membranes and the resulting improvements on the permeability and Intrinsic separation properties. It merits publication after revision on the following:

The authors need substantial help from a professional translator in order to improve English Language and style.

TGA diagrams must be included in the manuscript

Page 3 Lines 105-107: “Third, the MWCNTs-APTES contained hydrophilic amino groups and silicon ethoxy which can hydrolyze into hydrophilic hydroxyl group thus increased the hydrophilicity of the resulted TFN FO membranes.” After grafting MWCNTs-APTES contain amido groups.

Page 3 Line 128, 132: HNO3 and H2SO4, SOCl2 (Subscripts)

Page 3 Line 139 “The resultant solution was cast on the glass, and then immersed in DI water coagulation bath.” I m not sure I understand. Was the solution cast on a glass substrate (specifications?) and then (immediately?) the coated glass was immersed to the bath?

Page 4 Lines 208-209 “the characteristic peaks of 1050-1100 cm-1 and 1370-1400 cm-1 corresponded to stretching and bending vibrations of C-N and Si-O-C respectively” Not respectively 1050-1100 cm-1 is the Si-O-C stretching and at the 1370-1500 region the secondary amide bands.

Page 9 Lines 290-291 “It can be attributed to that the silicon ethoxy (groups) on the MWCNTs-PTES was (were) hydrolyzed into hydrophilic hydroxyl and amino groups.” Only hydoxyl.

Page 9 Lines 294-297 “The reason may that in the synthetic process of the PA layer, too much MWCNTs-APTES tended to agglomerate and heaped up on the TFN membrane surface, giving rise to the increased roughness of TFN membranes and reducing the hydrophilicity of TFN membranes” There is a strong possibility the increase in the hydrophobicity to be due to the formation of siloxane bridges. This can be proven by comparison of the IR spectra of the membranes.

Page 10 Lines 312-313 “In addition, the MWCNTs-APTES contained silicon ethoxy led to improve the hydrophilic of TFN membrane.” Actually it is the silanols (Si-OH) from the hydrolysis of ethoxy-silane groups that improve the hydrophilicity.

Author Response

Reviewers’ comments:

Reviewer #2

The present work describes a method (i.e. functionalization with aminopropyl triethoxy silane) to efficiently incorporate multi-walled carbon nanotubes to thin film forward osmosis membranes and the resulting improvements on the permeability and Intrinsic separation properties. It merits publication after revision on the following:

  1. The authors need substantial help from a professional translator in order to improve English Language and style.

Answer: Thank you for your suggestions. The English language and style have been improved in the revised manuscripts.

  1. TGA diagrams must be included in the manuscript

Answer: Thank you for your suggestions. TGA diagrams have been added in the Supporting Information (Figure S2).

Figure S2 TGA curves of MWCNTs, MWCNTs-COOH, MWCNTs-COCl and MWCNTs-APTES.

  1. Page 3 Lines 105-107: “Third, the MWCNTs-APTES contained hydrophilic amino groups and silicon ethoxy which can hydrolyze into hydrophilic hydroxyl group thus increased the hydrophilicity of the resulted TFN FO membranes.” After grafting MWCNTs-APTES contain amido groups.

Answer: Thank you for your suggestions. We have corrected it in the revised manuscripts.

  1. Page 3 Line 128, 132: HNO3 and H2SO4, SOCl2 (Subscripts)

Answer: Thank you for your suggestions. We have corrected it in the revised manuscripts.

  1. Page 3 Line 139 “The resultant solution was cast on the glass, and then immersed in DI water coagulation bath.” I m not sure I understand. Was the solution cast on a glass substrate (specifications?) and then (immediately?) the coated glass was immersed to the bath?

Answer: Thank you for your suggestions. The solution was cast on a glass substrate and then the coated glass was immersed into the bath.

  1. Page 4 Lines 208-209 “the characteristic peaks of 1050-1100 cm-1 and 1370-1400 cm-1 corresponded to stretching and bending vibrations of C-N and Si-O-C respectively” Not respectively 1050-1100 cm-1 is the Si-O-C stretching and at the 1370-1500 region the secondary amide bands.

Answer: Thank you for your suggestions. We have corrected it in the revised manuscripts.

  1. Page 9 Lines 290-291 “It can be attributed to that the silicon ethoxy (groups) on the MWCNTs-PTES was (were) hydrolyzed into hydrophilic hydroxyl and amino groups.” Only hydoxyl.

Answer: In the experiment, the amino group and the remained carboxyl group of the MWCNTs-APTES were hydrophilic, and the ethoxy silane on the MWCNTs-APTES was hydrolyzed into hydrophilic hydroxyl, which made the TFN membranes have better hydrophilicity.

  1. Page 9 Lines 294-297 “The reason may that in the synthetic process of the PA layer, too much MWCNTs-APTES tended to agglomerate and heaped up on the TFN membrane surface, giving rise to the increased roughness of TFN membranes and reducing the hydrophilicity of TFN membranes” There is a strong possibility the increase in the hydrophobicity to be due to the formation of siloxane bridges. This can be proven by comparison of the IR spectra of the membranes.

Answer: If there was the formation of siloxane bridges, the hydrophilicity of all the TFN membranes should be poorer than that of TFC membrane. However, all the water contact angles of TFN membranes were smaller than that of TFC membrane, showing that incorporating increased the hydrophilicity of the TFN membrane. For TFN-0.4 membrane, the water contact angle was smaller than that of TFN-0.2 membrane, therefore, we think that MWCNTs-APTES tended to agglomerate, increasing roughness of TFN-0.4 membrane and reducing the hydrophilicity of TFN-0.4 membrane.

  1. Page 10 Lines 312-313 “In addition, the MWCNTs-APTES contained silicon ethoxy led to improve the hydrophilic of TFN membrane.” Actually it is the silanols (Si-OH) from the hydrolysis of ethoxy-silane groups that improve the hydrophilicity.

Answer: Thank you for your suggestions. We have corrected it in the revised manuscripts.

Thank you for all your comments and suggestions!

06/16/2021

Reviewer 3 Report

Dear Authors

Congratulations for this good paper. For improvement some comments and suggestions are given in the attached manuscript.

Best regards,

The Reviewer

Author Response

Reviewers’ comments:

Reviewer #3

  1. Manuscript title is too long. If possible it should be shorter.

Answer: Thank you for your comments. We have made the title shorter as “Novel thin film nanocomposite forward osmosis membranes prepared by organic phase controlled interfacial polymerization with functional multi-walled carbon nanotubes”.

  1. resulting?

Answer: We have corrected it in the revised manuscripts.

  1. One figure indicating the flow direction, area, etc. can help the readers.

Answer: Thank you for your suggestions. Schematic diagram of cross-flow RO system has been added in the Supporting Information.

  1. Explain differences between A and Am.

Answer: A represents the pure water permeability, and Am represents the effective membrane area of the cross-flow FO equipment.

  1. Is this equation correct?

Answer: This equation is correct.

  1. Explain better these equations. Parameters B and S.

Answer: S is the membrane structural parameter, B is the salt permeability coefficient.

  1. Authors must include one color image of the membranes at a macroscopic scale.

Answer: Thank you for your suggestions. The color image of the TFN membranes at a macroscopic scale was added in Figure 6.

Figure 6. SEM images of surfaces and cross-section for PSF membrane (a, b), TFC membrane (c, d) and TFN-0.1 membrane (e, f).

  1. Cite the values of the angle for each case.

Answer: Thank you for your suggestions. The values of the angle for each case have been added in Figure 7.

Figure 7. Water contact angle of TFC and TFN membranes.

  1. Define all parameters in this table. See Equations 1, 3, 6 and 7. The same symbols?

Answer: Thank you for your suggestions. All parameters in this table have been defined, and there are the same as Equations 1, 3, 6 and 7.

Thank you for all your comments and suggestions!

06/16/2021
